# Characterization of the Complete Chloroplast Genome of the Dragonhead Herb, *Dracocephalum heterophyllum* (Lamiaceae), and Comparative Analyses with Related Species

Gui Fu [1,2], Yuping Liu [3,4,5], Marcos A. Caraballo-Ortiz [6], Changyuan Zheng [4], Tao Liu [1,4,5], Yujie Xu [1,4,5] and Xu Su [1,3,4,5,*]

1 School of Geosciences, Qinghai Normal University, Xining 810008, China; qhmdfg@163.com (G.F.); 201947341006@stu.qhu.edu.cn (T.L.); Eryaxu2018@163.com (Y.X.)
2 College of Ecological Environmental and Resources, Qinghai Minzu University, Xining 810007, China
3 Academy of Plateau Science and Sustainability, Qinghai Normal University, Xining 810016, China; lyp8527970@126.com
4 School of Life Sciences, Qinghai Normal University, Xining 810008, China; 15753735790@163.com
5 Key Laboratory of Medicinal Animal and Plant Resources of the Qinghai-Tibet Plateau in Qinghai Province, Qinghai Normal University, Xining 810008, China
6 Department of Botany, National Museum of Natural History, Smithsonian Institution, Washington, DC 20013-7012, USA; CaraballoM@si.edu
* Correspondence: xusu8527972@126.com

**Abstract:** *Dracocephalum heterophyllum* (Lamiaceae: tribe Mentheae) is an annual aromatic herb native to East Asia with a long record of human uses, including medicinal, alimentary, and ornamental values. However, no information is available about its molecular biology, and no genomic study has been performed on *D. heterophyllum*. Here, we report the complete chloroplast (cp) genome of *D. heterophyllum* and a series of comparative genomic analyses between this and closely related species of Lamiaceae. Results indicated that the cp genome has a typical circular structure of 150,869 bp in length, consisting of a long single-copy (LSC) region with 82,410 bp, a short single-copy (SSC) region with 17,098 bp, and two inverted repeat (IR) regions of 51,350 bp. A total of 133 genes were identified, including 37 tRNA genes, 8 rRNA genes and 88 protein-coding genes, with a GC content of 37.8%. The gene content, organization, and GC values observed here were similar to those of other *Dracocephalum* species. We detected 99 different simple sequence repeat loci, and the codon usage analysis revealed a preferential use of the Leu codon with an A/U ending. Comparative analysis of cp genome sequences revealed five highly variable regions with remarkably higher Pi values (>0.03). The mean $Ka/Ks$ between *D. heterophyllum* and three other *Dracocephalum* species ranged from 0.01079 (*psbB*) to 1.0497 (*ycf*2). Two cp genes, *ycf*2 and *rps11*, were proven to have high ratios of $Ka/Ks$, implying that cp genes may had undergone positive selection in the evolutionary history. We performed multiple sequence alignments using the cp genome of 22 species and constructed maximum likelihood (ML) and Bayesian trees, and found that *D. heterophyllum* were more closely related to *D. moldavica* and *D. palmatum*. In addition, the phylogenetic relationships between *Dracocephalum* and other members of Lamiaceae were consistent with previous results. These results are valuable for further formulating effective strategies of conservation and management for species in *Dracocephalum*, as well as providing a foundation for future research on the genetic resources of *Dracocephalum*.

**Keywords:** *Dracocephalum heterophyllum*; chloroplast genome; genomic structure; comparative analysis; phylogenetic analysis

## 1. Introduction

The mint family, Lamiaceae, comprise more than 7000 species from about 236 genera and seven subfamilies distributed worldwide [1]. Plants in this family include numerous herbs with medicinal and ornamental values, such as *Agastache rugosa*, *Mentha canadensisis*,

*Perilla frutescens*, and *Scutellaria baicalensis* [2]. The genus *Dracocephalum*, commonly known as "Dragonheads", comprises over 60 species of aromatic perennial herbs related to mints (tribe Mentheae) distributed in the northern temperate parts of the world. In China, 32 species and seven varieties of Dracocephalum are reported, mainly distributed in Northwest China [3]. Dragonheads have a long record in human history as medicinal herbs, and have been part of the traditional Tibetan and Uyghur medicines for centuries [4]. Previous studies have reported that the main active component found in *D. heterophyllum* is effective calming the mind, conferring protection for hypoxic brain damage, is antibacterial, and can alleviate diseases such as high blood pressure, lymphadenitis, and cough [4,5]. Moreover, a previous study by Zhang et al. has shown that essential oils of *D. heterophyllum* possess antimicrobial and antioxidant properties, and therefore, can be used as natural preservatives for food, cosmetics, and in pharmaceutics [6]. *D. heterophyllum* can better satisfy food needs of most livestock with a high feeding value. Meanwhile, it is usually developed as an auxiliary bee plant with advantage of much honey production and less powder [7].

In angiosperm plants, the chloroplast is a kind of essential plastid with a characteristic of matrilineal inheritance. As a place in which photosynthesis is conducted, it plays a vital role in many biochemical pathways, such as the metabolism of starch, fatty acids, nitrogen, amino acids, and internal redox signals [8]. Chloroplast is considered a semi-autonomous genetic organelle with an independent transcription and transport system [9]. The cp genome usually encodes 110–130 genes which varies in size from 75 to 250 kb [10,11], and which is divided into four parts: a long single-copy sequence (LSC, 80–90 kb), a short single-copy (SSC, 16–27 kb) sequence, and two reverse complement copies (IR, 20–28 kb) [12]. The frequent hybridization and variation of plants have made nuclear genomes are highly complex, thus, it is difficult to identify the orthologous genes [13]. In addition, evolutionary and phylogenetic analyses based on complete cp sequences can provide a greater amount of potential information of a higher quality than that obtained by analysis of one or more gene loci [14]. The existing studies have confirmed that cp gene order, gene content, and genome organization are highly conserved in plants [15]. Due to their conserved nature and comparatively high substitution rate, the cp genomes are widely applied in studies of species delimitation, systematics and evolution, and genetic engineering [16,17]. Since the complete cp genome of tobacco was sequenced successfully in 1986 [18], and the development of next-generation sequencing, more and more cp genome of plants has been sequenced. According to incomplete statistics, the complete cp genome sequences of more than 1100 plants have been collected in the NCBI GenBank database [19]. These data provide a fresh approach for phylogenetic analyses, genetic diversity evaluation, and molecular identification of plants [20–23].

*D. heterophyllum* has rich pharmacodynamic ingredients and some unique functions, which are worthy of development. In this contribution, we sequenced and analyzed the complete chloroplast genomes of *D. heterophyllum* using the Illumina NovaSeq6000 platform. We also downloaded the cp genomes of other *Dracocephalum* species from public databases. Our main goals were to characterize their structure, further perform genomic comparative analyses on these, and construct a phylogenetic tree, using whole cp genome sequences of *Dracocephalum* and the closely related species of Lamiaceae, to assess variations among their plastomes and confirm the evolutionary position of *D. heterophyllum*. Our results have multiple applications and can be used as a robust backbone for developing molecular markers to study the diversity of *Dracocephalum* chloroplast genomes, and to better understand the evolutionary relationships within this family. The findings of this research will provide guidance for the screening of quality germplasms and the design of conservation strategies for wild populations of *D. heterophyllum*.

## 2. Materials and Methods

### 2.1. Sampling and DNA Extraction

Fresh and healthy leaves of *D. heterophyllum* were collected in Gangcha County, Qinghai Province, China, at 3315 m elevation (37.20639°, 99.67556°). Leaves were rapidly stored in silica until dried. The specimens were deposited in the Herbarium of the Northwest Institute of Plateau Biology (HNWP), Chinese Academy of Sciences, Xining, Qinghai Province, China. The accession number is QTP-LJQ-CHNQ-026-1003. The total genomic DNA was then extracted from dried leaves using a modified CTAB method [24], and evaluated for quality and concentration using a NanoDrop 2000 spectrophotometer (Thermo Fisher Scientific, Waltham, MA, USA).

### 2.2. Chloroplast Genome Sequencing and Assembly

The DNA samples for testing were fragmented into 300~500 bp randomly by an ultrasonic processor (Covaris, Woburn, MA, USA). Then, we followed these steps: end repair and phosphorylate, add A-tailing, ligate index adapter, amplify and so on. A paired-end library of 2 × 150 bp with an insert size of ~350 bp was constructed and sequenced through the Illumina NovaSeq6000 platform. A total of 8,324,187 bp (2.5 Gb) of raw reads were obtained. After low-quality reads and reads with adaptors were filtered and trimmed using Fastp program [25], approximately 2.43 Gb of clean reads remained for the *D. heterophyllum* sample. The cp genome was assembled from the high-quality clean reads by performing NOVOplasty2.7.2 [26] with default Kmer 33 using the cp genome of *D. tanguticum* (MT457746) as a reference and the *rbcL* coding sequence (NCBI accession number NC_000932.1) of *Arabidopsis thaliana* as a seed sequence.

### 2.3. Annotation and Analysis of Chloroplast Genomes

The online program GeSeq [27] (https://chlorobox.mpimp-golm.mpg.de/geseq.html, accessed on 28 September 2021) and PGA software [28] were used to annotate the cp genome sequence. We compared annotations from the two measures and made final adjustments manually in Geneious version 11.0.2 [29]. We checked the initial annotation, putative starts, stops, and intron positions through comparison with homologous genes in the homogenera species *D. tanguticum*. Employing the program OGDRAW v1.3.1 [30], we drew a circular map of the cp genome of *D. heterophyllum* for the feature visualized. The cp genome sequence of *D. heterophyllum* was submitted to GenBank (Accession number OM201748).

### 2.4. Single Sequence Repeats (SSR) and Relative Synonymous Codon Usage (RSCU) Analysis

The online MISA program [31] was employed to detect SSRs using the following thresholds: ten, five, three, three, three, and three repeat units for mono-nucleotide, di-nucleotide, tri-nucleotide, tetra-nucleotide, penta-nucleotide, and hexa-nucleotide SSRs, respectively. We also conducted a relative synonymous codon usage (RSCU) analysis based on the sequence of protein-coding genes (PCGs), to reveal variations between synonymous and non-synonymous codon usage, regardless of the composition of amino acids, using CodonW (v1.4.2; https://downloads.fyxm.net/CodonW-76666.html, accessed on 14 January 2022) with default settings and length over 200 bp.

### 2.5. Complete Chloroplast Genome of Comparison Analysis

Boundaries among the four main chloroplast regions (LSC/IRb/SSC/IRa) of *D. heterophyllum* and six other related species of Lamiaceae were compared using IRscope [32]. In addition, *D. heterophyllum* was compared with three different species of *Dracocephalum* (*D. moldavica*, *D. palmatum*, and *D. tanguticum*) using mVista with the shuffe-LAGAN Mode [33]. The nucleotide diversity of four species in *Dracocephalum* was calculated through sliding window analysis in DnaSP (version 5.1) [34] with a window length of 600 bp and a step size of 200 bp.

*2.6. Analysis of Synonymous (Ks) and Non-Synonymous (Ka) Substitution Rate*

To analyze synonymous ($Ks$) and non-synonymous ($Ka$) substitution rates, the cp genome sequence of *D. heterophyllum* was compared with four other species of *Dracocephalum*. Here, we selected and extracted 79 exons from PCGs shared by four species for synonymous ($Ks$) and non-synonymous ($Ka$) substitution rates analysis. Then, sequences were aligned by MAFFT (version 7) [35], and substitution rates per exon were estimated using DnaSP software [36]. Coding genes were detected based on the following comparisons: *D. heterophyllum* vs. *D. moldavica*; *D. heterophyllum* vs. *D. palmatum*; and *D. heterophyllum* vs. *D. tanguticum*.

*2.7. Phylogenetic Analysis*

To examine the phylogenetic position of *D. heterophyllum* within Lamiaceae, an evolutionary tree was constructed, based on 22 full cp genomes sequences from nine genera in Lamiaceae downloaded from NCBI, using *Plantago depressa* as an outgroup (Table S1). Meanwhile, a molecular phylogenetic tree was constructed using 79 shared protein-coding genes of 21 cp genomes, with the exception of *Elsholtzia splendens*. All sequences of cp genomes and shared protein-coding genes were aligned with MAFFT (version 7) [35], and phylogenetic analyses were conducted according to the maximum likelihood (ML) method under the best-fit substitution model GTR + F + G4 selected by ModelFinder [37], using IQ-TREE (version 1.6.11) [38]. The bootstrap probability of each branch was calculated using 1000 replications. Additionally, a Bayesian inference (BI) analysis was conducted by MrBayes (version 3.1.2) [39], based on 22 full cp genomes sequences with the same best-fit substitution model. Four Markov chain Monte Carlo (MCMC) analyses were run simultaneously for two million generations, sampling one tree every 1000 generations with 25 percent of trees discarded as burn-in. The remaining trees were used to construct a consensus tree and to calculate the posterior probability (PP) percentages of trees.

## 3. Results

*3.1. General Characterization of Chloroplast Genome*

The assembled and annotated cp genome of *D. heterophyllum* was 150,869 bp in length and presented a typical tetra-partite circular structure with a pair of inverted repeats (IRs) of 51,350 bp each, which made the genome separate into two single-copy regions: a long single-copy (LSC) of 82,421 bp and short single-copy (SSC) of 17,098 bp (Figure 1). The estimated GC content was 37.8%, which was similar to the values published in congeners (Table 1). GC content for each of the three main regions of the cp (IR, LSC, and SSC) was calculated to be about 43, 36, and 32%, respectively, which is congruent with previous reports in related species (Table 1).

**Table 1.** Characteristics of chloroplast genomes for five species of *Dracocephalum* (Lamiaceae).

| Taxon | *D. heterophyllum* | *D. tanguticum* | *D. moldavica* | *D. palmatum* | *D. taliense* |
|---|---|---|---|---|---|
| GenBank accession | OM201748 | MT457746 | MT457747 | NC_031874.1 | MT473756 |
| Size (bp) | 150,869 | 150,594 | 149,868 | 150,510 | 150,976 |
| GC content (%) | 37.8 | 37.8 | 37.8 | 37.8 | 37.8 |
| IR (bp) | 51,350 | 51,370 | 51,352 | 50,772 | 51,358 |
| GC content in IR (%) | 43.2 | 43.0 | 43.0 | 43.1 | 43.0 |
| LSC (bp) | 82,421 | 82,221 | 81,450 | 82,160 | 82,253 |
| GC content in LSC (%) | 35.7 | 35.8 | 35.8 | 35.8 | 35.8 |
| SSC (bp) | 17,098 | 17,363 | 17,066 | 17,254 | 17,365 |
| GC content in SSC (%) | 31.8 | 31.9 | 31.8 | 31.8 | 31.6 |
| Gene number | 133 | 133 | 132 | 125 | 134 |
| tRNA gene number | 37 | 37 | 37 | 29 | 37 |
| rRNA gene number | 8 | 8 | 8 | 8 | 8 |
| Protein-coding gene | 88 | 88 | 87 | 88 | 89 |

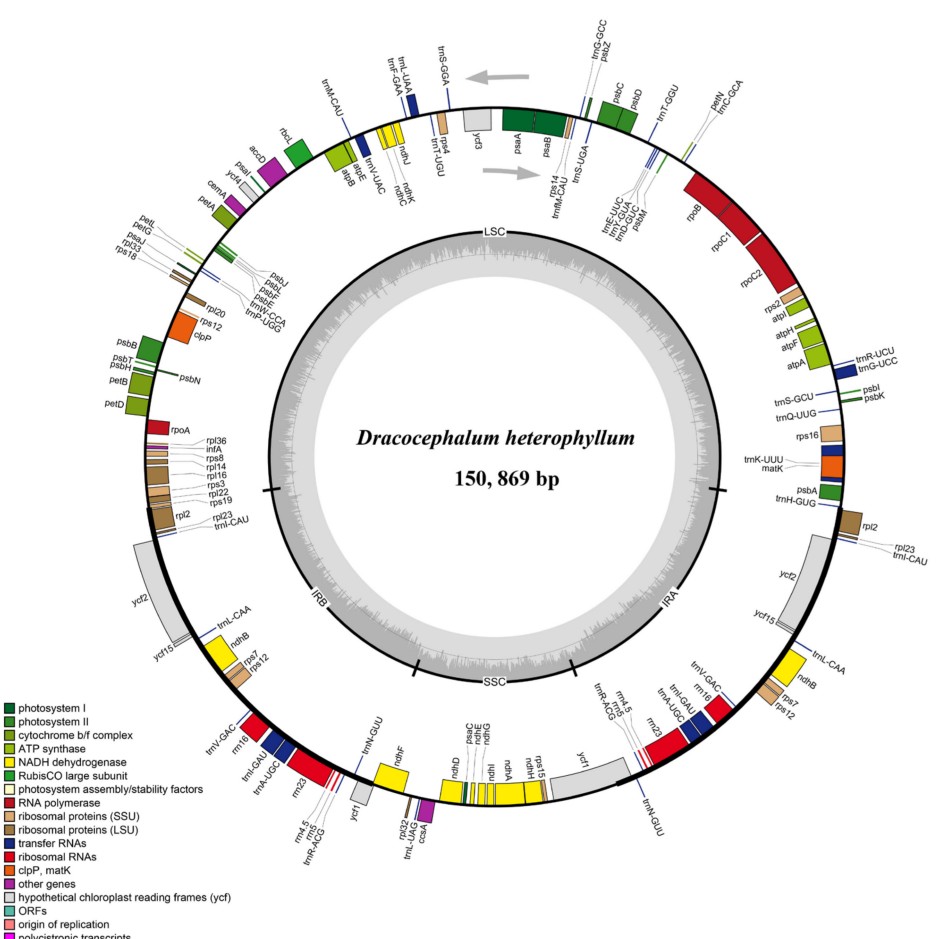

**Figure 1.** Gene map of the cp genome of *Dracocephalum heterophyllum* (Lamiaceae). Genes on the outside and inside of the circle represent the directions of transcribed in clockwise and counterclockwise, respectively.

A total of 133 genes, including 37 tRNA genes, eight rRNA genes, and 87 pro-tein-coding genes, were successfully annotated in the cp genome of *D. heterophyllum* (Table 2). The number of genes predicted for *D. heterophyllum* was higher than those for *D. palmatum* (125 genes). The distribution of protein-coding sequences in the cp ge-nome of *D. heterophyllum* reflects that 85 genes were located in the LSC region, 14 genes were in the SSC region, and 17 were duplicated in the IR regions. A total of 16 genes encoded one intron, including six tRNA and ten protein-coding genes, while two genes (clpP and ycf3) contained two introns, as previously reported for other Dracocephalum species. Among the 18 genes, the smallest intron was found in the trnL-UAA gene, be-ing 495 bp in length, whereas the largest one was in trnK-UUU, with 2535 bp and con-taining the matK gene. We present a comparison of the structure and size of genes with introns in D. heterophyllum and closely related species in Table S2.

### 3.2. Analysis of Simple Sequence Repeat (SSR) and Codon Usage

A total of 90 SSRs were found in the cp genome of *D. heterophyllum*, including six types of SSR, of which tri-nucleotide repeats were the most common (43, 47.7%), followed by mono-nucleotide (29, 32.2%) and compound-nucleotide (10, 11.1%) repeats (Table S3). Three additional types of SSRs were less abundant: di-nucleotide (three, 3.3%), tetra-nucleotide (three, 3.3%), and penta-nucleotide (two, 2.2%) repeats (Table S3, Figure S1). A detailed analysis of the frequencies of SSRs per genomic feature found that most were distributed in protein-coding (42) and intergenic (IGS; 36) regions, while ten of them were located in introns and two crossed the protein-coding-IGS boundaries. In addition, we also assessed

the distribution of 90 SSRs located in the LSC, IRb, SSC, and IRa regions, which were estimated to be 59 (65.6%), 10 (11.1%), 11 (12.2%), and 10 (11.1%), respectively (Table S4, Figure 2). We also obtained 88, 91, and 87 SSRs in the cp genome of *D. tanguticum*, *D. moldavica*, and *D. palmatum*, respectively (Table S5). They all included six types of SSR; tri-nucleotide repeats were the most common (48.2, 53.8, 55.1%), and no hexa-nucleotide SSRs were found (Figure 3, Table S5). A few tetra-nucleotides (2.2, 3.2, 2.2%) was detected in all three species, and only one penta-nucleotide (1.1%) and six di-nucleotides (6.8%) were found in *D. moldavica* and *D. palmatum*, respectively (Table S5). When combined, the number, which consisted of units, and distribution of SSRs in *Dracocephalum* showed high consistency.

**Table 2.** Annotated list of genes present in the chloroplast genome of *Dracocephalum heterophyllum* (Lamiaceae).

| Category | Group of Gene | Gene Name |
|---|---|---|
| Gene for photosynthesis | Subunits of photosystem I | *psaA, psaB, psaC, psaI, psaJ* |
| | Subunits of photosystem II | *psbA, psbB, psbC, psbD, psbE, psbF, psbH, psbI, psbJ, psbK, psbL, psbM, psbN, psbT, psbZ* |
| | Subunits of NADH dehydrogenase | *ndhA, ndhB* [1] (2×), *ndhC, ndhD, ndhE, ndhF, ndhG, ndhH, ndhI, ndhJ, ndhK* |
| | Subunits of cytochrome b/f complex | *petA, petB* [1], *petD* [1], *petG, petL, petN* |
| | Subunits of ATP synthase | *atpA, atpB, atpE, atpF* [1], *atpH, atpI* |
| | Large subunit of rubisco | *rbcL* |
| Self-replication | Proteins of large ribosomal subunit | *rpl2* [1] (2×), *rpl14, rpl16* [1], *rpl20, rpl22, rpl23, rpl32, rpl33, rpl36* |
| | Proteins of small ribosomal subunit | *rps2, rps3, rps4, rps7* (2×), *rps8, rps11, rps12* (2×), *rps14, rps15, rps16* [1], *rps18, rps19* |
| | Subunits of RNA polymerase Ribosomal RNA | *rrn4.5* (2×), *rrn5* (2×), *rrn16* (2×), *rrn23* (2×) |
| | DNA-dependent RNA polymerase | *rpoA, rpoB, rpoC1* [1], *rpoC* [2] |
| | Transfer RNA | *trnA-UGC* [1] (2×), *trnC-GCA, trnD-GUC, trnE-UUC, trnF-GAA, trnG-GCC, trnG-UCC* [1], *trnH-GUG, trnI-CAU* (2×), *trnI-GAU* [1] (2×), *trnK-UUU* [1], *trnL-CAA* (2×), *trnL-UAG, trnL-UAA* [1], *trnfM-CAU, trnM-CAU, trnN-GUU* (2×), *trnP-UGG, trnQ-UUG, trnR-ACG* (2×), *trnR-UCU, trnS-GCU, trnS-GGA, trnS-UGA, trnT-GGU, trnT-UGU, trnV-GAC* (2×), *trnV-UAC* [1], *trnW-CCA, trnY-GUA* |
| Biosynthesis | Maturase | *matK* |
| | Protease | *clpP* [2] |
| | Envelope membrane protein | *cemA* |
| | Subunit Acetyl-CoA carboxylase | *accD* |
| | C-type cytochrome synthesis gene | *ccsA* |
| | Translation initiation factor | *infA* |
| Unknown | Conserved open reading frame | *ycf1* (2×), *ycf2* (2×), *ycf3* [2], *ycf4, ycf15* (2×) |

Note: [1] Gene containing a single intron; [2] Gene containing two introns; 2× showed genes duplicate.

We detected the codon usage frequency for the cp genome of *D. heterophyllum* and three closely related species based on the sequence of protein-coding genes (CDS), whose length were 85,371, 80,154, 79,269, and 79,599 bp. A total of 28,457 codons and RSCU were estimated, comprising 64 different types, which included 20 amino acids and three stop codons in *D. heterophyllum* (Figure 4, Table S6), which were similar to the other three species. In all species, leucine was the most abundant amino acid with 2982, 2864, 2840, and 2843 instances and encoded by six codons, accounting for 10.5, 10.7, 10.7, and 10.7% of the total (Table S7). Both methionine (ATG) and tryptophan (UGG) had only one codon type with 656, 644, 642, 635, and 489, 475, 470, 468 instances, respectively, and presented no bias (RSCU = 1.00) (Table S7). The codon UUA for leucine with the highest RSCU values were 1.9 (943 instances, 33.1%), 1.89 (893, 33.8%), and 1.87 (886, 33.4%) in *D. heterophyllum*,

*D. moldavica*, and *D. palmatum*, respectively (Table S7). All species had identical codons (AGC) for serine with the lowest RSCU values. Except for *D. heterophyllum* (31), 30 codons had a RSCU value of greater than one, representing 18,014 (67.4%), 17,786 (67.3%), and 17,869 (67.3%) codons in three closely related species, respectively (Table S7). Remarkably, all codons with values higher than one ended with A or U, except for *trnL-CAA* and *trnS-GGA* (Table S7).

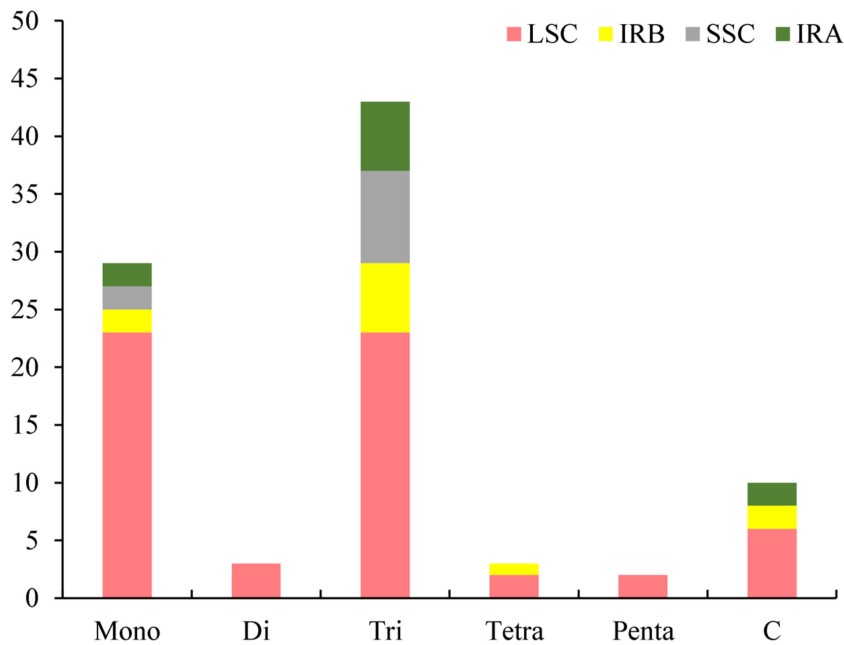

**Figure 2.** Distribution of different types of SSRs located in different regions of the cp genome of *Dracocephalum heterophyllum* (Lamiaceae). Colors represent different regions.

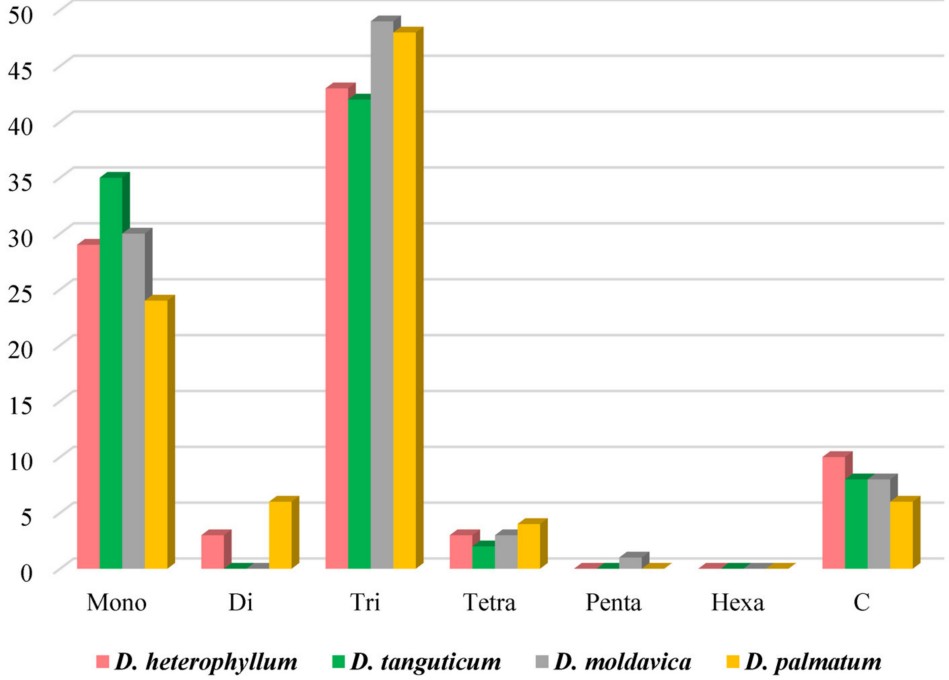

**Figure 3.** The type and presence of simple sequence repeats (SSRs) in the cp genome of four species in *Dracocephalum*. Colors represent different species.

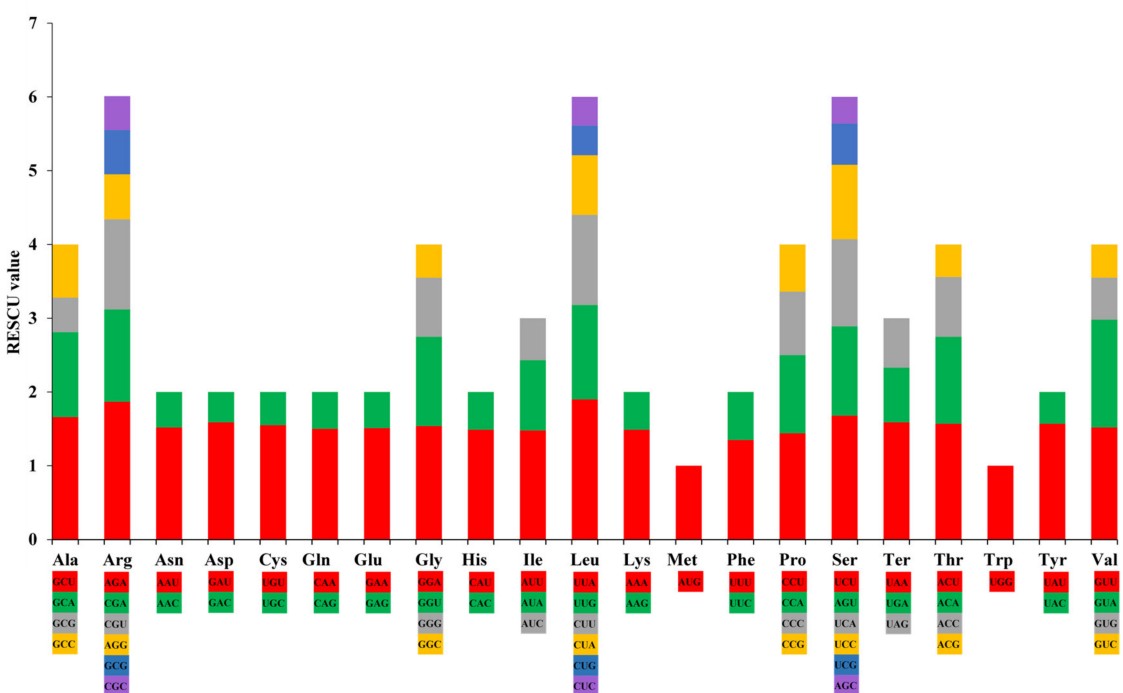

**Figure 4.** RSCU analysis of 21 amino acids and three termination codons in *Dracocephalum hetero-phyllum* (Lamiaceae). Bar diagram in different colors reflects codon usage bias (red, green, grey, yellow, cyan, and purple correspond to the proportion of different codons in descending order), while numbers represent codon quantities.

### 3.3. IR Contraction and Expansion

We compared the cp genome of *D. heterophyllum* with six closely related species in Lamiaceae, including two congeners (*D. moldavica*, *D. tanguticum*) and four other genera (*Elsholtiza densa*, *Mentha spicata*, *Ocimum basilicum*, and *Perilla frutescens* var. *hirtella*). A detailed comparison was performed for four genomic boundaries (LSC/IRa, LSC/IRb, SSC/Ira, and SSC/IRb) of the seven species (Figure 5). Our results showed that the lengths of LSC, IR, and SSC regions were similar among the genomes of Lamiaceae. While the IR organization was highly conserved, ranging from 50,234 to 51,394 bp, there were minor variances with expansions and contractions. Both *psbA* and *rpl22* genes were entirely located within the LSC region, and the *rpl2* gene was entirely located within the IR organization. The *rps19* gene of *E. densa* was utterly inside the LSC region, but in the other six species it was positioned at the boundary between the LSC and the IRb. The pseudogene *ycf1* and the gene *ndhF* were also positioned at the junctions of the IRa/SSC and IRb/SSC regions, respectively and their sequences showed length variabilities among species. Notably, the *ndhF* gene of three species in Dragonhead, and *M. spicata*, *P. frutescens* var. *hirtella*, overlapped with *ycf1* in the IRb region. The *trnH* gene was consistently placed at the IRa/LSC border in all species except for *M. spicata*, which was contained far inside the LSC region. Instead, *M. spicata* was the only species with the *rps19* gene positioned near the IRa/LSC border.

### 3.4. Comparisons of the cp Genome between D. heterophyllum and other Lamiaceae species

To identify events of genomic divergences among the eight species of Lamiaceae, we used the mVISTA program, with *Plantago depressa* as a reference (Figure 6). Results indicated that the IR region is more conserved than the LSC and SSC, especially for rRNA genes (*rrn4.5*, *rrn5*, *rrn16*, and *rrn23*). Also, the non-coding intergenic regions were highly divergent, especially for *atpH–atpI*, *ndhH–ndhA*, *petN–psbM*, *rbcL–accD*, *rrn5–trnL*, *trnI–rpl23*, *trnK–rps16*, *trnT–psbD*, *trnV–rps7*, and *trnV–rps12*. However, highly divergent regions were

also detected within protein-coding regions, such as in *accD*, *matK*, *ndhD*, *ndhF*, *ndhH*, *rpl22*, *rpoB*, *rpoC2*, *ycf1*, and *ycf2*.

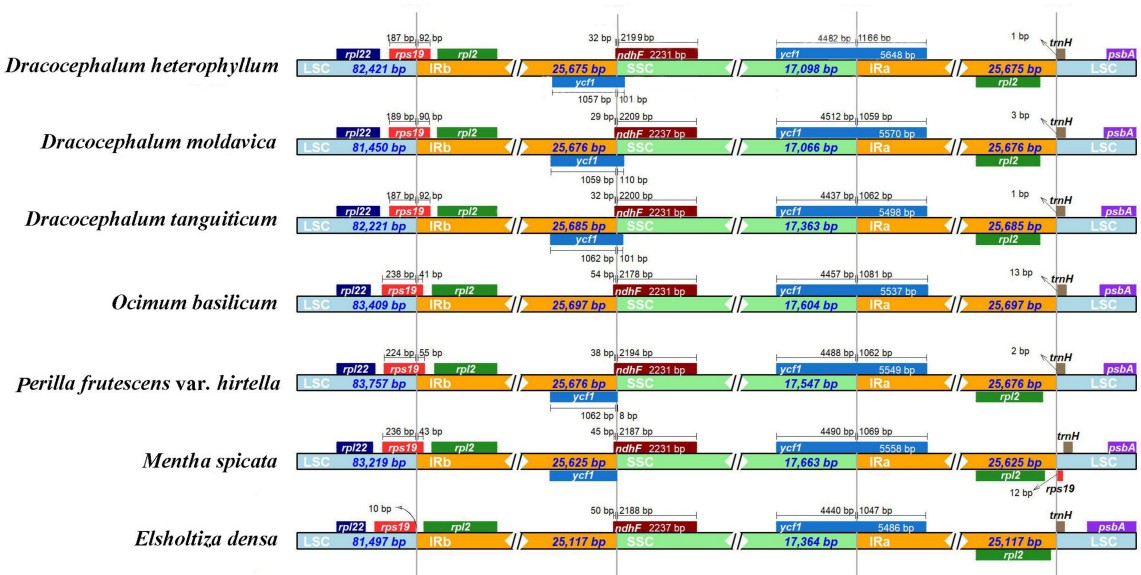

**Figure 5.** Comparison of the borders of the LSC, SSC, and IR regions of cp genomes between *Dracocephalum heterophyllum* and related species in Lamiaceae.

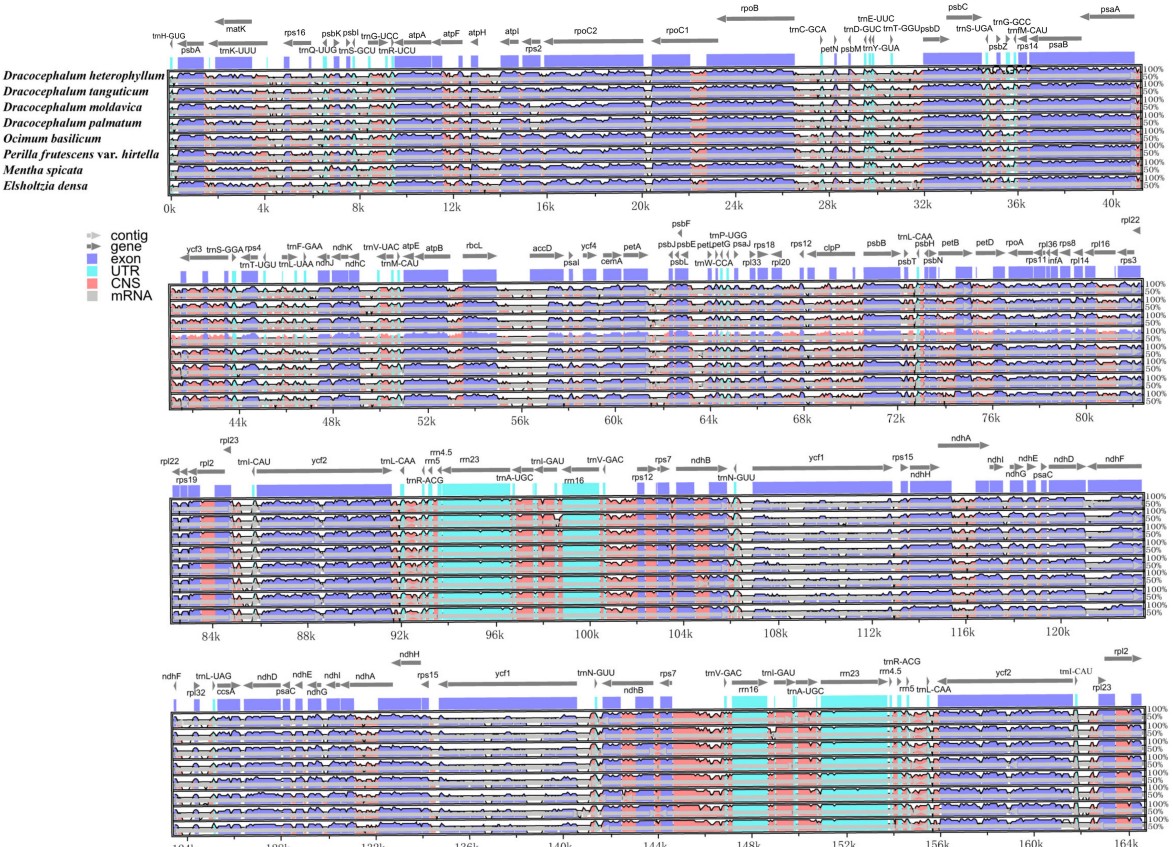

**Figure 6.** The sequence alignment map of eight cp genomes of Lamiaceae using *Plantago depressa* as a reference. The *Y*-axis indicates identity from 50 to 100%. Gray arrows indicate the position and direction of each gene. Purple indicates the exons of protein-coding genes (exon); Blue indicates the untranslated region (UTR); Red indicates conservative non-coding sequences (CNS).

In addition, we conducted a nucleotide diversity analysis for four species of *Draco-cephalum* (*D. heterophyllum*, *D. tanguticum*, *D. moldavica,* and *D. palmatum*) in DnaSP, which identified a total of 2958 mutations (Eta), 2918 polymorphic sites (S), and an average nucleotide diversity value of 0.01075 (Pi) (Table S8). When compared with the values obtained for the LSC (0.01357) and SSC (0.01509) regions, the IR showed a lower variability (0.00447), and therefore, it was the most conserved region in the cp genome. Five mutational genomic hotspots were detected with remarkably higher Pi values (>0.03), including three intergenic spacers (*rps16–trnQ*, *trnT–psbD*, *ndhF–rpl32*) and two conserved open reading frames with unknown functions (*ycf3*, *ycf1*). From these, three regions (*rps16-trnQ*, *trnT-psbD*, *ycf3*) were found in the LSC region, while the other two (*ndhF–rpl32*, *ycf1*) were located in the SSC segment (Figure 7).

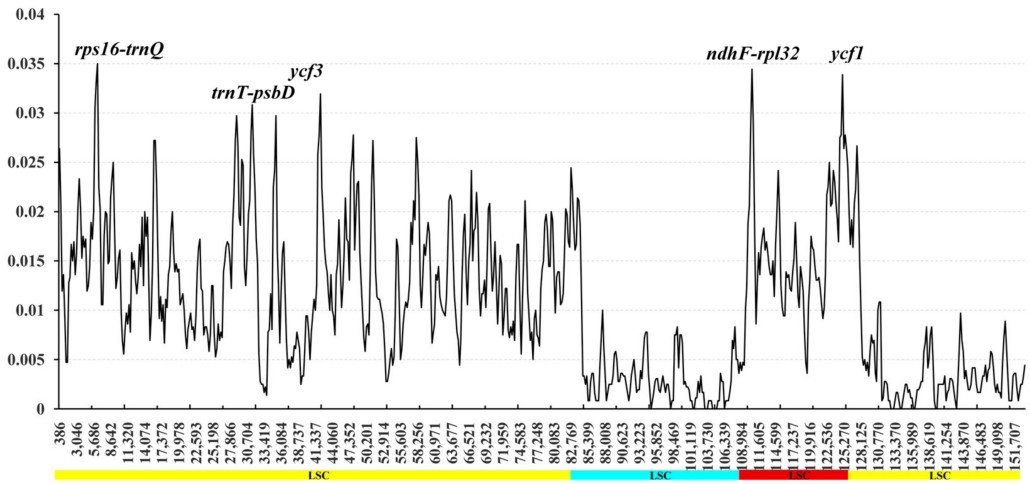

**Figure 7.** Nucleotide diversity within cp genomes of four species in *Dracocephalum* using a sliding window analysis. Window lengths: 600 bp; step size: 200 bp. Five hypervariable genes or gene spacers with the highest Pi values (>0.03) are highlighted. *X*-axis: position of the midpoint of a window; *Y*-axis: nucleotide diversity of each window.

### 3.5. Characterization of Substitution Rates

Ratios of non-synonymous (*Ka*) and synonymous (*Ks*) substitutions for 79 PCGs were calculated by comparing *D. heterophyllum* with three congeners (*D. heterophyllum* vs. *D. moldavica*, *D. heterophyllum* vs. *D. palmatum*, and *D. heterophyllum* vs. *D. tanguticum*). Thirty-one of the PCGs (*atpB*, *atpH*, *ndhB*, *ndhE*, *petG*, *petN*, *psaC*, *psaJ*, *psbA*, *psbC*, *psbE*, *psbF*, *psbI*, *psbJ*, *psbK*, *psbH*, *psbL*, *psbM*, *psbN*, *psbT*, *psbZ*, *rpl14*, *rpl23*, *rpl36*, *rps7*, *rps14*, *rps15*, *rps16*, *rps18*, *ycf3*, *ycf15*) showed values of zero, indicating that no synonymous or non-synonymous changes occurred. In the residual 48 PCGs, results showed that the mean *Ka/Ks* ratio between *D. heterophyllum* and three other *Dracocephalum* species ranged from 0.01 (*psbB*) to 1.05 (*ycf2*) (Table S9). However, the Ka/Ks ratio for most genes was less than one, suggesting that they experienced negative selection, except for *rps11* and *ycf2*, who displayed positive selection (*Ka/Ks* > 1) (Figure 8).

### 3.6. Evolutionary and Phylogenetic Analysis

Phylogenetic trees were constructed based on whole cpDNA sequences of 22 species from subfamilies Lamioideae (2), Nepetoideae (18), and Scutellarioideae (2) in Lamiaceae, using *Plantago depressa* (Plantaginaceae) as an outgroup. The ML and Bayesian trees showed two major monophyletic clades within Lamiaceae, receiving strong support from both bootstraps and Bayesian posterior probabilities (BP = 100%; PP = 1, respectively; Figure 9). The phylogenetic trees constructed with complete cp genome and CDS sequences have the same topology (Figure S2). Within the tree, taxa from the same genus were clustered together. The *D. heterophyllum* was sister to a clade consisting of *D. moldavica* and *D. palmatum* and was clustered together with *D. tanguticum*. The largest monophyletic clade was

for the subfamily Nepenthoideae, which included 16 species from six genera distributed in three tribes, most of them strongly supported. Tribe Mentheae comprised three genera, with *Dracocephalum* being a sister to *Mentha* with *Salvia* at the base of this relationship. The second subclade in Nepetoideae formed by Elsholtzieae and Ocimeae received moderate statistical support (BP = 52%; PP = 0.741). However, *Elsholtzia* and *Perilla* were strongly supported as sisters (both in tribe Elsholtzieae), while *Ocimum* was monophyletic (tribe Ocimeae). Meanwhile, subfamilies Lamioideae and Scutellarioideae were strongly supported as being sisters, despite lacking resolution with respect to their relationship with Nepetoideae. Genera within each of these two subfamilies (i.e., *Pogostemon*, *Stachys,* and *Scutellaria*) were monophyletic, and the overall structure of the phylogeny is in agreement with current classification schemes of Lamiales.

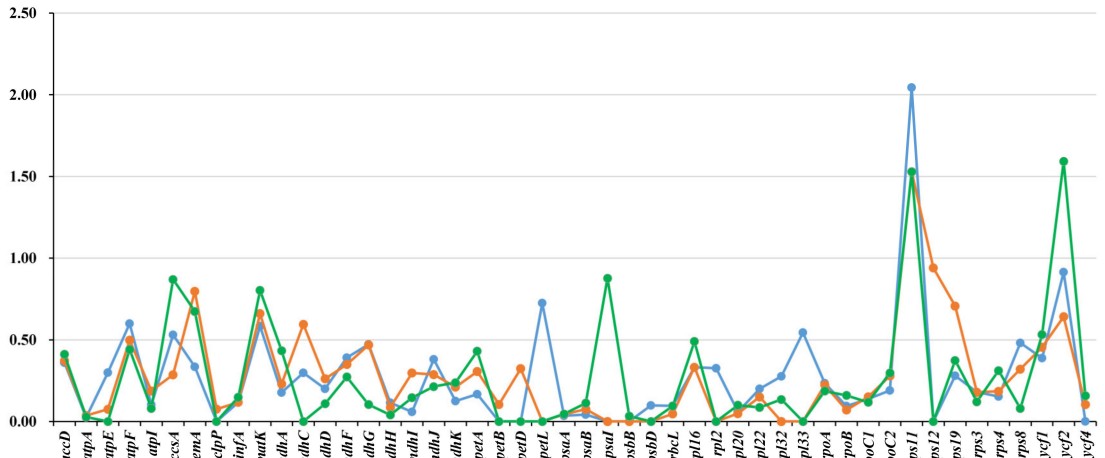

**Figure 8.** Comparison of the ratio of non-synonymous (*Ka*) to synonymous (*Ks*) substitutions of 79 protein-coding genes among the cp genomes of *Dracocephalum heterophyllum* vs. *D. moldavica*, *D. palmatum*, and *D. tanguticum*. *Y*-axis: Value of *Ka/Ks*; *X*-axis: gene names. Blue lines indicate the comparison *D. heterophyllum* vs. *D. moldavica*; orange lines are *D. heterophyllum* vs. *D. palmatum*; and green lines *D. heterophyllum* vs. *D. tanguticum*.

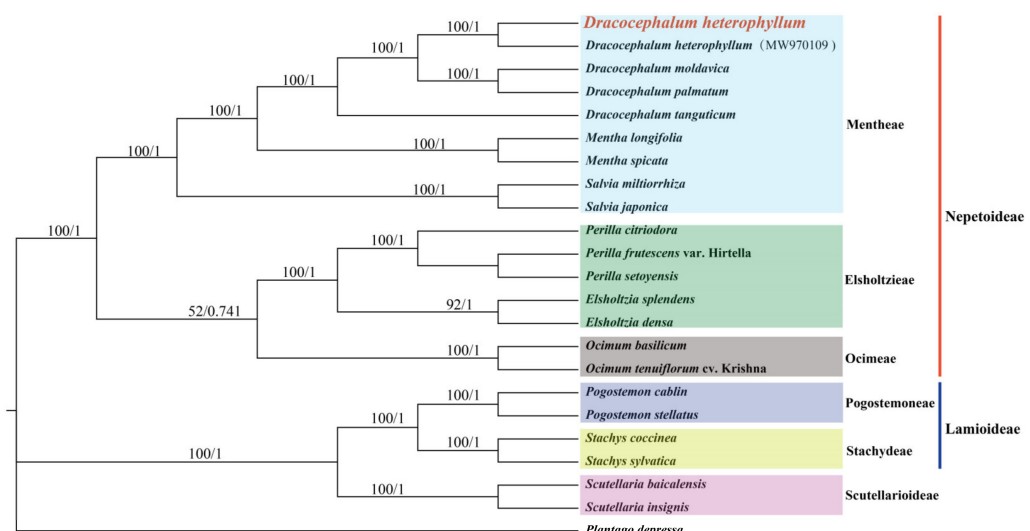

**Figure 9.** Cladogram based on the cp genomes of 22 species of Lamiaceae. Numbers at nodes represent bootstrap and Bayesian posterior probabilities (BP/PP). Different colors represent currently recognized subfamilies and tribes within Lamiaceae.

## 4. Discussion

### 4.1. Organization and Features of cp Genomes

Our study examined the features, content, and organization of the cp genome of *D. heterophyllum* and compared it with the cp genomes of four other species of *Dracocephalum* that have been previously published, showing that all of them presented the typical quadripartite structure found in vascular plants [40]. The cp genomes of the five *Dracocephalum* species ranged from 149,868 (in *D. moldavica*) to 150,976 bp (in *D. taliense*), indicating that they are very conserved, only exhibiting minor differences that altered their sizes. Intriguingly, the cp genomes of *Dracocephalum* had similar sizes as those of distant species within Lamiaceae, such as *Elsholtzia densa* [41] (149,095 bp), *Stachys byzantine,* [42] (149,749) and *Scutellaria baicalensis* [43] (151,817 bp). The phenomenon of minimal differences in cp size within a genus and a family has been previously reported in the genera *Sorghum* (Poaceae) [44] and *Ilex* (Aquifoliaceae) [45], and in the subfamily *Coryloideae* of Betulaceae [46], among others. The size variations of cp genomes above are usually the result of the expansions of the IR regions during evolution [47].

Regarding GC content, the overall amount detected in the cp genome of *D. heterophyllus* is 37.8%, identical to the values reported for the four other congeners evaluated. The GC content within the IRs region (43.2%) was higher than the estimate for the other two regions (LSC, 35.7%; SSC, 31.8%), which might be related to the presence of high GC nucleotide percentages in the genes *rrn4.5*, *rrn5*, *rrn16*, and *rrn23*, as previously reported [8,48,49].

Regarding gene estimates, our analysis detected 133 genes in the cp genome of *D. heterophyllum*, including 37 tRNA, 8 rRNA, and 88 protein-coding genes. The number and arrangement of genes were generally consistent when compared to those of *D. moldavica*, *D. taliense*, and *D. tanguticum*. However, some differences among them were detected. Gene number (133 genes) and cp genome arrangement between *D. heterophyllum* and *D. tanguticum* were identical, but comparisons with the other two species showed minor differences: *D. moldavica* had 132 genes, due the loss of *rps2*, while *D. taliense* contained 134 genes because of a duplication of *rps19*. In fact, the duplicated *rps19* pseudogene has also been reported in other species of Lamiaceae [50,51]. However, there have been no reports for *rps2* missing in other species of Lamiaceae. Thus, these minor variations of gene content in the cp genome of *Dracocephalum* were caused by evolutionary events of gene deletions and insertions. Additionally, we found that there was specific trans-splicing in the *rps12* gene of *D. heterophyllum*, where the 5′ end of the exon was located within the LSC region while the 3′ end was in the IR region, which is a situation commonly observed in many angiosperms [52].

Although introns do not encode proteins, they play a critical role in regulating gene expression [53]. The cp genomes of the earliest diverging angiosperms contained a complete repertoire of 18 genes with introns [54]. Within the cp genome of *D. heterophyllum*, those 18 genes with their introns also were detected, of which 16 contained one, and two contained two introns. Although the number and arrangement of genes with introns in the cp genomes of several *Dracocephalum* species were identical and their exons highly conserved, we detected considerable variation in intron length, especially in genes such as *clpP, ndhA,* and *ycf3*. Intron polymorphisms (IP), including length polymorphism (ILP) and single nucleotide polymorphism (ISNP), can be used as efficient molecular markers, which have been previously developed and applied for analyses of multiple lineages of plants [55].

### 4.2. Simple Sequence Repeats (SSRs) and Codon Usage Analysis

The variations among SSR copy numbers are distinctive among plant species, and therefore, provide an important source of genomic information suitable for phylogenetic studies [56,57]. Our study analyzed the number and distribution of different SSR motifs in the cp genome of *D. heterophyllum* and closely related species. The number, motif type, frequency, and distribution for detected SSRs showed high uniformity in the cp genomes of the four species of *Dracocephalum*. Among the SSRs, tri-nucleotide repeats were the richest, and more than half of SSRs were located in non-coding spacers (NCS), suggesting that

these regions have the potential to be hotspots for mutations. Our results suggest that SSRs can be used to develop DNA markers with the potential to delimit species, reconstruct phylogenetic relationships, and assess genetic diversity in *Dracocephalum* [58].

Previous studies have revealed that codon usage has essential functions in expressing genetic information, and therefore, it is critical for the evolution of cp genomes [47,59]. Here, a total of 28,457 codons were detected in the CDS of *D. heterophyllum*, the amino acids encoding leucine (10.5%) being the most common, and the least common being the ones for cysteine (1.2%), which is consistent with findings in *D. moldavica*, *D. tanguticum*, and *D. palmatum* (Table S6). It has been reported that species with close phylogenetic relationships may adopt similar codon selection strategies [60]. In the case of *Dracocephalum*, most amino acids had codon bias with a high preference (RSCU > 1), except for methionine and tryptophan (RSCU = 1). The RSCU value of codon types was more significant than one ending in A or U, except for *trnL-CAA* and *trnS-GGA*. Research have shown that the codons with the highest AT-content in the cp genomes of terrestrial plants preferentially end with A/U, and this might be one of the causes why these codons are more prevalent in dicotyledons [61–63]. Therefore, these findings may contribute to further understanding of the evolutionary history of *Dracocephalum*, especially through natural selection and mutation pressures [40].

### 4.3. IR Expansion/Contraction of cp Genomes

Variations in the size of cp genomes of angiosperms are usually caused by expansions and contractions at the IR/SC borders [64]. Li et al. [65] suggested that the length of the IR regions in some species of Magnoliaceae have a positive correlation with the total length of the cp genome sequence. In our study, a comparison of the SC/IR boundaries among the seven cp genomes of Lamilaes revealed that the gene orders were identical, although there was some minor variation in the IR regions. Notably, the *ycf1* gene, considered a pseudogene, was partially duplicated in all species except for *O. basilicum* and *E. densa*. Our results showed that the downstream sequences of IRb/SSC were conserved and the *ndhF* gene was adjacent to the IRb/SSC borders, which were consistent with the general pattern described for angiosperms [66]. The gene content and orders of SC/IR boundaries in three species of *Dracocephalum* were identical, suggesting that boundary region between the SSC and the two IR was relatively conserved. Last, the comparative genome analysis results with mVISTA revealed that the *D. heterophyllus* cp genome was relatively conserved, presenting some minor variation mainly localized at non-coding regions, due to insertions and deletions [67].

### 4.4. Comparative Genomes and Characterization of Substitution Rates

Although the cp genomes of land angiosperms are highly conserved, mutational hotspots in cp genomes are usually conserved at the generic level, which can be used to obtain multiple informative loci suitable for DNA barcoding and phylogenetic research [20]. Results obtained from mVISTA for cp genome divergence analysis in Lamiaceae suggest that *Dracocephalum* show low degrees of sequence divergence, with plastomes being relatively conserved. However, divergence hotspots for conserved non-coding sequences (CNS) regions and gene protein-coding regions were detected among all eight genomes of Lamiaceae. Similar phenomena have been observed in the cp genomes of other genera such as *Scutellaria* [43], *Physalis* [59], and *Zygophyllum* [40]. Meanwhile, we also identified five highly variable regions with high Pi values (>0.03) according to the sliding window analysis, including three intergenic sections (*rps16–trnQ*, *trnT–psbD*, and *ndhF–rpl32*) and two genes with unknown function and conserved open reading frames (*ycf1* and *ycf3*). Previous multispecies studies found that intergenic spacers can serve as markers with high resolution for phylogenies [46]. For example, *rps16–trnQ* are highly variable in most plants and have been used for DNA barcoding in phylogenetic studies of 12 different genera across angiosperms [20]. Also, the gene *ycf1* is at more variable loci suitable for barcodes of land plants than existing candidate barcodes [68]. Therefore, these highly variable regions

in the cp genome of *Dracocephalum* are expected to provide enough genetic information to conduct studies on species delimitation and the phyletic evolution of Lamiaceae.

Synonymous (*K*s) and non-synonymous (*K*a) nucleotide substitution patterns are considered to be essential parameters for gene evolution analysis [69], and the *K*a/*K*s ratio can account for the selection pressure of genes [19,70]. Three directions of evolution were revealed through the value of *K*a/*K*s, where genes underwent purifying (<1), neutral (=1), and positive (>1) selections [70]. In our studies, the *K*a/*K*s value of the remaining 31 PCGs could not be calculated because their *K*a or *K*s was zero, suggesting that these genes were conserved without any non-synonymous or synonymous nucleotide substitutions. The values of *K*a/*K*s for 48 PCGs were all less than one except for *rps11* and *ycf2*, indicating that the majority of PCGs of the four *Dracocephalum* samples experienced purifying selection. These results were consistent with previous reports that non-synonymous substitutions are less frequent than synonymous ones, and *K*a/*K*s ratios for most PCGs were less than one [71]. Given that purifying selection could sweep away deleterious mutations, as the primary pattern in natural selection [72], we inferred that most genes in the cp genomes of *Dracocephalum* underwent extensive purifying selection to sustain their conserved functions, and the evolutionary rate was relatively slow. This result correlates with previous analyses showing that the cp genome of *D. heterophyllus* is relatively conserved and with minor variation, mainly localized at non-coding regions.

### 4.5. Phylogenetic Analysis Based on cp Genomes

In the past three decades, many authors have focused on the phylogenetic relationships within Lamiaceae and have established a framework for the systematics and evolution of Lamiaceae [73–75]. Cp genome sequences have been successfully used to reconstruct phylogenetic relationships among plant lineages [76], and a phylogenetic tree of Lamiaceae using cp genomes could be used to better understand the evolution of this diverse family. Two data sets (the complete cp genome sequences and concatenated protein-coding sequences) recovered the same topology, which was congruent with previous trees constructed using molecular data for Lamiaceae. There are two monophyletic clades within the family, forming six groups corresponding to the six currently described tribes. The first monophyletic clade consisted of the tribes Mentheae, Elsholtzieae, and Ocimeae from subfamily Nepetoideae, while the second clade constituted the tribes Scutellarioideae, Stachydeae, and Pogostemoneae from the subfamily Lamioideae, where the former is sister to the latter two. Notably, the systematic relationships among the three tribes in Nepetoideae remain ambiguous [73]. Previous studies have shown three inconsistent relationships: (1) Ocimeae being sister to the Mentheae–Elsholtzieae branch [77], (2) Mentheae being sister to the Ocimeae–Elsholtzieae group [78–81], and (3) Elsholtzieae being sister to the Mentheae–Ocimeae clade [82]. Our result, however, supported the second relationship, where Mentheae is sister to the Ocimeae–Elsholtzieae branch [73], although with weak nodal support (BP = 52, PP = 0.74). Hence, more studies are still needed to resolve relationships among these three tribes. Overall, we consider that the phylogenetic relationships between *Dracocephalum* and other genera has been resolved through our studies, confirming previous findings within Lamioideae from Li et al. [1] and Zhao et al. [73]. Therefore, we propose that whole cp genomes could provide sufficient data information to reconstruct the phylogenetic relationship of plants, particularly within the family Lamiaceae.

### 5. Conclusions

In the present study, we sequenced the cp genome sequences of the Dragonhead herb *Dracocephalum heterophyllum* (Lamiaceae). After comparing this newly sequenced cp genome with closely related species, we found that the cp genome size, GC content, and gene number and order among species was found to be highly conserved. The location and distribution of 99 repeat sequences and some highly variable regions were identified. We expect that these variable regions and repeat markers can assist future studies on species delimitation, systematics and evolution, and genetic engineering of

*Dracocephalum*. The mean *Ka/K*s between *D. heterophyllum* and three other *Dracocephalum* species ranges from 0.01 (*psbB*) to 1.05 (*ycf2*). Two genes, *ycf2* and *rps11*, were found to have high *Ka/K*s ratios, implying that they may have undergone positive selection during their evolutionary history. Phylogenetic trees constructed using whole cp genomes and protein-coding sequences support *D. heterophyllum* as being a member of the tribe Mentheae in the subfamily Nepetoideae. The results obtained in this study are expected to provide valuable genetic resources to develop strategies to discern species, perform molecular breeding, and assess shallow relationships among *Dracocephalum* species in the future.

**Supplementary Materials:** The following supporting information can be downloaded at: https://www.mdpi.com/article/10.3390/d14020110/s1, Figure S1: Number of different forms of SSRs identified in the cp genome of *Dracocephalum heterophyllum* (Lamiaceae). Colors represent different nucleotide repeats; Figure S2: Phylogenetic trees of the *Dracocephalum* species based on the sequences of cp protein-coding gene by ML; Table S1: Information of reported chloroplast genomes that was used for phylogenetic trees constructing; Table S2: Length of genes with introns in the cp genomes of *D. heterophyllum*, *D. moldavica*, *D. tanguticum,* and *D. taliense*; Table S3: Characteristics and distribution of 90 simple sequence repeats (SSRs) in the chloroplast genome of *Dracocephalum heterophyllum* (Lamiaceae); Table S4: Statistics of identified simple sequence repeats in the chloroplast genome of *Dracocephalum heterophyllum* (Lamiaceae); Table S5: SSR analysis for the cp genome of four species in *Dracocephalum*; Table S6: The codon usage frequency for the cp genome of *D. heterophyllum*; Table S7: The comparison of codon usage frequency for the cp genome of four species in *Dracocephalum*; Table S8: Nucleotide diversity analysis for four species of *Dracocephalum* in DnaSP; Table S9: The *Ka/K*s value of 48 PCGs in three compares (*D. heterophyllum* vs. *D. moldavica*, *D. heterophyllum* vs. *D. palmatum,* and *D. heterophyllum* vs. *D. tanguticum*).

**Author Contributions:** The experimental design was completed by X.S., Y.L. and G.F.; Samples collection and treatment were conducted by C.Z., T.L. and G.F.; G.F., C.Z., T.L. and Y.X. led data analysis, and figures and tables with assistance from C.Z. and T.L.; The manuscript was drafted by G.F. and X.S., and M.A.C.-O. edited the manuscript for structure, language, and scientific content. All authors have read and agreed to the published version of the manuscript.

**Funding:** This research was funded by [the Second Tibetan Plateau Scientific Expedition and Research (STEP) program] grant number [2019QZKK0502]; [the Science and Technology Project of Qinghai Minzu University] [2021XJGH14]; [the Key Laboratory of Medicinal Animal and Plant Resources of the Qinghai Tibet Plateau in Qinghai Province] grant number [2020-ZJ-Y40].

**Institutional Review Board Statement:** Not applicable.

**Data Availability Statement:** The original contributions presented in the present study are publicly available, and accession numbers can be found Table S1. The datasets generated for this study can be found in GenBank, the accession number of final plastome is OM201748. The associated BioProject ID, Bio-Sample accession and SRA are PRJNA797531, SAMN25008769 and SRR17629219 for raw sequencing data, respectively.

**Conflicts of Interest:** The authors declare no conflict of interest.

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
