# Peer review of "Characterization of the Complete Chloroplast Genome of the Dragonhead Herb, Dracocephalumheterophyllum (Lamiaceae), and Comparative Analyses with Related Species"

_diversity, doi:10.3390/d14020110_

Round 1

Reviewer 1 Report

  1. As you mentioned, the genus Dracocephalum comprise over 60 species, so why did you only sequenced and assembled the only one cp genome of this genus? Is there anything special about heterophyllum? What is the significance of your research?
  2. The significance of the research is vague, lacking a clear purpose and hypothesis. The author once mentioned the development of DNA barcodes, but I can't find its purpose from the article. Is there a problem of unclear species classification in this genus? The introduction section must be re conceived and written.
  3. The bioinformatic streamline was previously used in similar kind of researches but on different subject. Besides, the complete chloroplast genome of heterophyllum had been published in NCBI (GenBank: MW970109.1) with the different authors Zhang,L., Wang,Z. and Wei,Y. (https://www.ncbi.nlm.nih.gov/nuccore/2047403206). So can you provide your accession number? Is your study different from previous studies? Why reassemble when the chloroplast genome has been published by others? From this point of view, your research hardly provides new experimental and original data. Moreover, the raw sequencing data should be submitted, too.
  4. The reference genome used in assembly (Dracocephalum palmatum) and annotation (D. tanguticum) is different. Why?
  5. The genomes used in many analyses are not uniform. I'm confused about the author's purpose. For example, in line 135, there were total eight species in the IRscope analysis according to your description, but there were actually only seven species in Figure 5. Moreover, in Figure6, according to common sense, the reference genome in this analysis is not displayed. If you use 8 sequences, the final result should be 7 bands, but there are 8 bands in your figure, which is very strange. And the quality of Figure 8 is too low. I can hardly get any information from it. In addition, I wonder why the chloroplast genomes of five species in this genus do not appear in any analysis at the same time. It seems that the author is deliberately avoiding something and selectively using some chloroplast genomes for some specific analysis. For example, D. taliense is missing in Figure 8 as well as only four species were used in nucleotide diversity analysis. It is too messy.
  6. sequence repeat (SSR) and codon usage only performed in heterophyllum. Lacking the comparison with other species.
  7. Is your phylogenetic tree constructed based on the full length of the chloroplast genome? Or protein coding genes? Or the shared protein coding genes? No relevant description was found in the method section.

Reviewer 2 Report

The authors sequenced the complete chloroplast genome of Dragonhead Herb, Dracocephalum heterophyllum (Lamiaceae) and compered it with the public data of related species. This manuscript provides valuable information for genetic improvement of the Dragonhead Herb, Dracocephalum heterophyllum (Lamiaceae). The work seems to be thorough and sound but I have some concerns about both the content and the presentation. Please see the comments below for further details.

Please see the following points of clarification that would greatly improve the readability of this work:

English language editing was not done; some of the sentences simply don't make sense

Abstract should be summarized only indicating the key information regarding objectives, methodology and obtained results and conclusions.

lines 83-96: In the Introduction section, Objectives must be clearly explained in a separate paragraph. Authors must summarize and restructure these objectives and the information showed in the lines 83-96

I recommend using some other relevant papers to enrich introduction and discussion. For example I recommend to address to the following papers:

Song, Y., Zhang, Y., Xu, J. et al. Characterization of the complete chloroplast genome sequence of Dalbergia species and its phylogenetic implications. Sci Rep 9, 20401 (2019). https://doi.org/10.1038/s41598-019-56727-x

Arab MM, Marrano A, Abdollahi-Arpanahi R, Leslie CA, Askari H, Neale DB, Vahdati K. (2019) Genome-wide patterns of population structure and association mapping of nut-related traits in Persian walnut populations from Iran using the Axiom J. regia 700K SNP array. Scientific Reports 9(1): 6376.

Sadat-Hosseini M, Bakhtiarizadeh MR, Boroomand N, Tohidfar M, Vahdati, K. (2020). Combining independent de novo assemblies to optimize leaf transcriptome of Persian walnut. PloS One, 15(4), e0232005.

Sithichoke Tangphatsornruang, Pichahpuk Uthaipaisanwong, Duangjai Sangsrakru, Juntima Chanprasert, Thippawan Yoocha, Nukoon Jomchai, Somvong Tragoonrung (2011) Characterization of the complete chloroplast genome of Hevea brasiliensis reveals genome rearrangement, RNA editing sites and phylogenetic relationships. Gene,475, 2: 104-112.

Line 93 and 94: The relationship of the present study with the assessment of secondary metabolic regulation is not clear? Please correct this part

Line 100: Please complete the materials and methods section. The exact number of samples is not clear

Line 108: Chloroplast genome sequencing and assembly is incomplete. The steps of preparation sequencing library are not well detailed. Which one of the restriction enzymes is used? It is also important to refer to the original method which authors followed.

Line 113: Why the authors did not use BWA (Burrows Wheeler-aligner) for assembly? I would suggest to use different aligner and compared them with each other.

Line 114: Why did you use only one cp reference genome? I suggest using another reference genome as it will probably improve your findings

Line 122: How did you validate your annotation?

Line 141: There is a lack of connection between Analysis of synonymous (Ks) and non-synonymous (Ka) substitution rate and other analysis. This question must be clarified.

Results must be summarized; now they are a description of the obtained data but they are not focused in the important things that you found in your work. Discussion and conclusion. Again, the same problem, the relevance of the work is not shown. Too many references.

Data archiving and data availability. I suggest authors deposit cp genome sequence into NCBI that can be used for future research of the Dragonhead Herb, Dracocephalum heterophyllum (Lamiaceae)

Reviewer 3 Report

Line 114-15, as a seed is not clear.

You need to explain a little more how did you assemble the genome.

You can move figure 2 to a supplemental file or present it better. The quality of the presentation is low. 

For discussing SSR type (Table 3), please see this article: https://www.nature.com/articles/s41598-019-39793-z

Figures 3,4 need to be re-draw again. Find another way to present this data.

I believe you do not need to add all the figures in the main text; instead, move some of them to supplementary files. You completed all the appreciable cp analysis, but you do not need to include all the information in the article. Just focus on the central finding.

The results and discussion section are long. Make the results and discussion section shorter. Just focus on the main finding and avoid using extra information.  

Round 2

Reviewer 1 Report

According to my suggestions, the author made more detailed modifications and answered my questions carefully. The quality of the article has been improved and seems to have reached the publishing level, but there are still some small problems that need to be corrected.

1.Some language expressions need to be polished.

2.In addition to using whole cp genome sequence, please use the shared protein coding genes to construct the ML tree, too, in which the codon alignment method should be adopted. And the results could be compared with the existing results.
